# A Comprehensive Analysis of the UVC LEDs’ Applications and Decontamination Capability

**DOI:** 10.3390/ma15082854

**Published:** 2022-04-13

**Authors:** Talita Nicolau, Núbio Gomes Filho, Jorge Padrão, Andrea Zille

**Affiliations:** 12C2T—Centre for Textile Science and Technology, University of Minho, 4800-058 Guimaraes, Portugal; tali_nicolau@hotmail.com (T.N.); padraoj@2c2t.uminho.pt (J.P.); 2School of Economics and Management, University of Minho, 4710-057 Braga, Portugal; id7657@alunos.uminho.pt

**Keywords:** light-emitting diodes, LEDs, Ultraviolet C, UVC, decontamination capability, disinfection, sterilization, traditional light sources, light bulb lamps, systematic literature review

## Abstract

The application of light-emitting diodes (LEDs) has been gaining popularity over the last decades. LEDs have advantages compared to traditional light sources in terms of lifecycle, robustness, compactness, flexibility, and the absence of non-hazardous material. Combining these advantages with the possibility of emitting Ultraviolet C (UVC) makes LEDs serious candidates for light sources in decontamination systems. Nevertheless, it is unclear if they present better decontamination effectiveness than traditional mercury vapor lamps. Hence, this research uses a systematic literature review (SLR) to enlighten three aspects: (1) UVC LEDs’ application according to the field, (2) UVC LEDs’ application in terms of different biological indicators, and (3) the decontamination effectiveness of UVC LEDs in comparison to conventional lamps. UVC LEDs have spread across multiple areas, ranging from health applications to wastewater or food decontamination. The UVC LEDs’ decontamination effectiveness is as good as mercury vapor lamps. In some cases, LEDs even provide better results than conventional mercury vapor lamps. However, the increase in the targets’ complexity (e.g., multilayers or thicker individual layers) may reduce the UVC decontamination efficacy. Therefore, UVC LEDs still require considerable optimization. These findings are stimulating for developing industrial or final users’ applications.

## 1. Introduction

Light-emitting diodes (LEDs) combine negative and positive semiconducting materials to create band gaps where electrons flow in-between and reconnect at their junctions [1,2,3]. This semiconductor process creates a narrow-spectrum light, representing a critical difference from traditional light sources [4,5]. On the other hand, these conventional sources apply heat, ionized gas, or arc discharge to create light [4]. Like these sources, LEDs can provide a broad range of wavelengths varying from infrared to Ultraviolet C (UVC) [4]. The LED producers dope the diodes’ semiconductor single junctions with different materials [3,4,5] to generate this broad range. For instance, producers use aluminum gallium nitride in semiconductors to reach the UVC wavelength band [3].

Recently, almost 140 countries became parties in a global agreement (Minamata Convention on Mercury) that should reduce products containing mercury manufacture and trade [6,7,8,9], including mercury vapor lamps. These traditional sources of UVC have been widely applied for commercial decontamination purposes [10]. This preference is majorly due to its low cost and cost-effectiveness [11]. Nevertheless, this global agreement might decrease traditional mercury vapor lamps’ market share, increasing the demand for alternatives to this widespread application and enhancing the market penetration of LEDs. LEDs represent an exciting alternative to mercury vapor lamps when users want more robust, flexible, durable, and eco-friendly sources of UVC [12,13,14]. Researchers have been testing LEDs in decontamination systems. Besides the source selection, operators must also consider wavelength output. A typical selection of decontamination wavelength range corresponds to UVC (200–280 nm, [15]).

Nevertheless, not all sources of UVC emit the whole range. Low-pressure mercury vapor lamps, for example, peak at 254 nm, while UVC LEDs, which peak at 265 nm, offer the entire range. Figure 1 depicts this difference in terms of wavelengths. In addition to this difference, the image also points to the optimal wavelengths for achieving antimicrobial properties.

UVC displays remarkable antimicrobial properties toward different microorganisms [1,16,17,18,19]. The UV irradiation causes various photolesions on DNA strands (the photodimerization process, shown in Figure 2), being the [2 + 2] photoaddition of thymine bases the most common [20]. This photolesion stimulates consecutive DNA bases to bind. Once these abnormal binds happen, they hinder the nucleic acid transcripts elongation [8,10,19,21,22,23,24] and further impede other DNA functions. Furthermore, UV radiation induces biochemical processes (i.e., enzyme catalase or nitric oxide synthase) that lead to the creation of reactive oxygen species (ROS) [25]. These processes and the environment in the biochemical process happened to create different types of ROS, like superoxide anion ($O_{2}^{-}$), hydrogen peroxide ($H_{2}O_{2}$), or hydroxyl ion (${\mathrm{OH}}^{-}$) [3,6,25]. These species are responsible for protein oxidation [18,21,26], lipid peroxidation [25] and changes in gene expression [18].

Decontamination represents a broader concept, and users might need different levels depending on their scientific fields or the issue at hand [23]. Then, we can narrow this concept down into two additional concepts: disinfection and sterilization. These two concepts are crucial for users willing to reuse or increase the life cycle of products and materials. The disinfection process reduces microorganisms to a safe level for the user without eliminating spores, achieving at least three log-reduction of the microorganisms’ concentration [27]. While, sterilization processes inactivate microorganisms and spores to a nonviable level, resulting in at least a six log-reduction of the concentrations [27]. Therefore, sterilization guarantees the user safer levels than disinfection. It is established that UVC LEDs can decontaminate different targets. However, it is unclear to what extent. For instance, traditional UVC sources are recognized for their decontamination capability due to their irradiation intensity, reaching disinfection and sterilization levels [16]. On the other hand, the decontamination capability of LEDs remains unclear, especially for manufactured products and materials, in general.

Hence, this research uses a systematic literature review (SLR) to enlighten three aspects: (1) the UVC LEDs’ application according to the area, (2) the UVC LEDs’ application in terms of different biological indicators, and (3) the comparison between the UVC LEDs’ decontamination effectiveness and the conventional lights’ effectiveness. 

Recognizing each aspect increases the likelihood of applying UVC LEDs in decontamination settings, and they aim at users in different stages. The application in multiple areas broadens the comprehension of this decontamination method and this alternative light source for users already applying UVC LEDs. This step presents (dis)similarities among areas and provides ideas on new approaches to current problems. The analysis of biological indicators helps potential users already interested in applying UVC LEDs in their systems. These potential users may be more concerned with one type of indicator than with others or with the appearance of a new biological problem within their application. In parallel with this, we provide summarized results that serve as a benchmark for these potential users’ experiences. Finally, the comparison between LEDs and conventional lights looks at convincing unsure users of the application of UVC LEDs. These users might be looking for alternatives to their decontamination systems but might not be convinced about the LEDs possibility. Thus, this aspect groups a series of positive and negative features that might help them decide.

Furthermore, Section 2 presents the application of UVC LEDs in distinct scientific fields. Section 3 indicates the results of UVC LED for different biological representatives. Section 4 compares the characteristics between LEDs and other methods or UVC sources, including the decontamination effectiveness. Finally, Section 5 provides concluding remarks regarding the application of UVC LEDs. For readers interested in the SLR steps (Methods), results, and details, see Appendix A. Those interested in the comparison among the decontamination effectiveness of UVC LEDs and other methods applied within our SLR’s database, see Appendix B.

This review is very important as tool for making an informed decision about the use, or not, of UVC-LED in function of the type of material to be disinfected and the surrounding environment. The carefully analysis of the increasing use of UVC-LEDs for disinfection, their capability and when they are an alternative to conventional mercury-based UVC light lamps is of paramount importance on the light of the recent advance on LED technology. The pros and cons of LED technology was scrutinized for disinfection taking in account their sterilization efficacy, cost, type of material, and irradiation efficiency.

## 2. The Decontamination Action of UVC LEDs in Different Areas

This section divides the information obtained at our SLR for each scientific field, linking to the first part of this research objective. We provide only the critical information in this section. 

### 2.1. Human Health

Healthcare-associated infections (HAIs) represent a common health problem. This issue, though, does not only affect patients. It hits a whole set of stakeholders in the healthcare scenarios, such as direct and indirect workers and visitors. In this context, healthcare facilities apply decontamination methods to protect the stakeholders from HAIs (e.g., surgical site infection and infections associated with central-line or catheter [21,22,28,29,30]). The methods used to decontaminate utensils in healthcare settings may be physical or chemical [16]. Nevertheless, to properly define the best strategy applicable for this process, users must identify the forming materials on these tools. Otherwise, their typical lifecycle may be severely jeopardized. Among the physical methods, several settings use UVC light to decontaminate often-touched surfaces and equipment impossible to immerse [8].

The SLR’s results demonstrate a concentration of studies [3,8,17,21,22,23,28,29,30,31,32,33,34,35,36,37,38,39,40] in this scientific field. Of these studies, all found some level of decontamination effectiveness. In terms of disinfection (at least three log-reduction), we have seven studies [3,24,33,34,35,36,37,38] indicating it. Meanwhile, ten studies [8,17,21,23,28,29,30,31,32,39] indicate elimination of biological indicators (sterilization) or growth inhibition. In terms of lower levels of decontamination effectiveness, only one study [22] discussed it. Moreover, one additional study [40] evaluated the UVC LEDs’ efficiency with another method (optical density compared to the control).

Researchers have proposed solutions to stimulate UVC LEDs’ acceptance in this context. For instance, they have been minimizing these methods’ disturbance on users, using rapid exposures (i.e., 1-s [8,17,24] or 10-s [24,31]) or portable devices [22,28,33]. This ease of use represents an essential feature for stakeholders, especially workers. If processes are burdensome, they will prefer other disinfection options, or none, in the worst-case scenario. Another common feature of UVC LEDs researchers demonstrate is their effectiveness in repeatedly reprocessing equipment. For instance, Messina et al. [28] found significant decontamination effectiveness of UVC LEDs on stethoscopes after 243 h. Finally, in terms of features, researchers defend UVC LEDs application by comparing decontamination with other methods, finding results like the percentages ranges with other decontamination methods such as 0.5% chlorhexidine [23], alcohol [28], or distilled water [40]. Researchers have also discussed UVC LEDs parameters, like exposure time, the distance from the UVC source, and surface characteristics [22,28,31,32,33]. This discussion is crucial for quick reproducibility in the health context. 

For health purposes, UVC applications generate some common preoccupations. The first issue relates to the damage UVC may cause to DNA and its possible toxicity. Although this is a crucial concern, some studies [17,36,37] found no significant results related to neither DNA damage nor cell toxicity. Conversely, these results are dose-dependent when dealing with blood elements [21]. A second common concern is the penetration of deeper structures. Literature indicates this is a common problem for other UV types (UVA, 400–315 nm, or UVB, 315–280 nm [15]) [17,39], but not much for UVC. One must regard that UVB can penetrate epithelial tissue, reaching the inner epidermis layers. In contrast, UVA penetrates an even deeper layer, reaching the dermis layer [25].

### 2.2. Animal Health

Only one study within our database concentrated on Veterinary issues [41]. Romanchenko et al. [41] applied UVC LEDs in a hive to eliminate/reduce varroosis in bees. The importance of this application relates to the possibility of this mites’ infestation affecting food security [41]. Although there are multiple methods to fight this problem, each has its disadvantages [41]. According to the authors [41], the UVC application reduced from 67.2% to 83.86% of the mites following the evaluated years. This application produced additional benefits: (1) the reduction of dedicated work and (2) the prevention of environmental conditions that could reduce the number of bees in the hives (reduction of wasps and birds) [41].

It is important to remark that although our SLR just found the previously discussed research, others exist. For example, Moreno-Andrés et al. [42] applied UVC LEDs combined with chemicals or photocatalytic thin films to prevent bacteria proliferation in recirculation aquaculture systems. These researchers found synergistic results among the combination, and when they coupled thin films and UVC-LEDs, they achieved a 2-log reduction in common bacteria samples from these systems [42]. These results represented a 55% improvement compared to UVC LEDs’ results found by these authors [42]. 

### 2.3. Air Treatment

Some studies [7,43,44,45,46] focused on using UVC LEDs for air treatment. Few studies within our SLR database considered the decontamination effectiveness. Two studies [7,45] were able to disinfect some of their biological indicators but not all, while one research [46] achieved sterilization in part of its biological targets. The rest of the studies focused on different indicators [44] or did not reach the disinfection level [43]. A common concern researcher in this area has been airborne transmitted pathogenic infections, occurring either via droplets or aerosol [7,43,44,45,46], which is crucial during the current pandemic.

Due to the COVID-19 pandemic, there has been an increasing concern about air quality. COVID-19 spreads majorly via aerosols [32] emitted by infected persons. Actions like coughing, sneezing, and even talking may lead to infected aerosol released into the environment. Researchers have been proposing strategies to decontaminate closed environments to deal with this issue in the literature. Muramoto et al. [32] developed an air purifier that combines UVA/UVC-LEDs and a HEPA filter, a honeycomb ceramics filter, and a pre-filter. In this system, the LEDs are responsible for irradiating the surface of the HEPA filter to decontaminate microorganisms trapped in it. According to the authors, combining these tools led to the faster elimination of floating influenza viruses.

There is an association between UVC decontamination effectiveness and biological indicators taxonomies [7]. This finding may be decisive, given that it might determine UVC application. Moreover, there is a positive relation between decontamination effectiveness and airflow conditions [46]. In poorly ventilated environments, indoor air has lower convection leading to the environmental accumulation of pathogens, increasing the likelihood of infection [38]. Also, some researchers exemplify compact systems [43,44,46] that would be easy to implement in different settings to improve the applicability. 

Like researchers in health, researchers interested in air treatments are also interested in presenting UVC LEDs’ gains related to other decontamination methods. For instance, Lee et al. [45] found UVC possessed higher decontamination efficacy than ozone and failed to achieve synergistic effects. Nevertheless, users can combine different methods to promote synergistic effects, as Lee et al. [45] did and ultimately found.

### 2.4. Water Treatment

There is a special interest in water treatment [2,5,10,11,19,26,47,48,49,50,51,52,53,54,55,56,57,58,59,60,61]. Several reasons are at the core of this interest, such as people’s limited access to potable water and industrial applications [52,57]. Research interest covers drinking water [10,19,57], and rainwater [55] disinfection, wastewater [11,52,58,60] reuse, sewage [53] and food processing water [54] management. Besides the treatment, researchers are also concerned with the parameters that may improve or hinder UVC application. For example, substances turning into opaquer liquids would reduce UV treatment effectiveness at different levels [19,51]. Water circulation and exposure time improve water decontamination effectiveness [56]. Interestingly, water volume causes a dubious effect on UVC treatment, insignificant [56] or slightly better in lower volumes [57]. Moreover, the UVC irradiation types (continuous or pulsed) provide comparable results [49].

Depending on the water intended final application the treated water will have, there are different decontamination targets the treatments must present. For instance, if users aim to provide drinking water, the treatment should disinfect the surface with at least a 4-log reduction [19]. The reuse of wastewater, excreta, and greywater, on the other hand, only if treatments disinfect water (at least 3-log reduction) [52]. All the studies in our SLR found some decontamination levels in their results. Few studies [5,10,47,48,50,55] were unable to achieve disinfection levels, presenting biological reductions of less than 3-log. Most of the studies in this scientific field [2,11,26,47,49,52,53,56,57,58,60] accomplished disinfection levels, and three studies achieved sterilization levels [19,51,54,59]. Such results give confidence for users to apply UVC for its decontamination capabilities in the water.

The application of UV to disinfect water has been gaining supporters since the most common methods are chemical (i.e., chlorination and ozonation). However, these methods generate persistent residual carcinogenic by-products (such as chlorine or bromate) [5,11,49]. Furthermore, these methods have led to new resistant microorganisms [5,49] and affected the organoleptic properties of water [49]. Some researchers combined different UV wavelengths in their treatments [2,5,26,47,48,52], or they proposed a combination of UV treatments with other methods [11,58,61]. These combinations envisaged possible additive or synergistic effects, which were ultimately identified. For instance, the combination of UVA and UVC [5,26,48] or UVB and UVC [2], UVC with other light sources (like excilamps [61]), or UVC with chemical oxidants [58] lead to synergistic (or additive) effects.

Lastly, the application of UV on water treatments comprises a common preoccupation: the possible bacterial effectiveness of recovering from the UV decontamination effect. These microorganisms use the dark repair and the photoreactivation processes [2,20,40,43] to recover from the UVC impact. Both processes are further discussed in Section 3.2. 

### 2.5. Food Treatment

Food decontamination is another research interest for researchers applying UVC treatment [1,6,62,63,64,65,66,67,68,69]. Food treatment against pathogens has been a well-established process since the advent of pasteurization, which is equivalent to a disinfection process once it leaves spores intact. However, pasteurization is not applicable for all food items as it is a thermal method [62,67]; thus, different strategies are required. Furthermore, there has been an increase in the demand for fresh and ready-to-eat(use) products, which are proliferous media for foodborne pathogens [6,62,63,65,68].

The application of UVC to treat food gained traction in the last years because this method barely affects nutritional values or quality aspects of the food [62]. This concern seems crucial for some researchers, as they specifically evaluated possible changes in those parameters [63,64,65,66,68]. Interestingly, researchers have also compared different UVC irradiation types, either continuous or pulsed [14,60]. To provide final users an idea of a broad range of applications, researchers have been using UVC treatment for various types of food. For example, fruits [64,65], vegetables [62,63], raw fish [68] and meat [62], cooking ingredients [67], ice [59], sausages [6], mushrooms [6] and cheese [1]. 

Most studies provided decontamination effectiveness [62,63,64,65,66,67,68,69], two studies [1,6] reached disinfection levels, and one achieved sterilization [59]. Nevertheless, one must comprehend that such aggressive treatment as sterilization may significantly impact food nutritional values and quality in this type of application. 

Finally, the authors had also compared UVC treatment with traditional food decontamination methods like the application of slightly acidic electrolyzed water (SAEW) [63,65,66] or fumaric acid (FA) [65]. Furthermore, Lu et al. [69] compared different ultraviolet decontamination capabilities. These comparisons indicate UVC treatment generated better results than these traditional methods individually. However, UVC presents additive or synergistic effects [63,65,66,69] in those proposed combinations.

### 2.6. Materials

The last group from our analysis encompasses the treatment of materials [70,71,72]. Researchers were concerned with UVC decontamination effectiveness on different ball types [70], surfaces (e.g., carpet or laminate) [71], and food contact surfaces [72].

These different applications provide awareness for future users of UVC’s possible applications. According to these studies, UVC has shown some decontamination effectiveness [72], but most likely disinfection levels [70,71]. In addition to the decontamination effectiveness, Wood et al. [71] evaluated relative humidity impact on the decontamination effectiveness and compared conventional UVC sources with LEDs.

Just like it happened for foods, Trivellin et al. [70] tested whether the UVC irradiation caused any visual changes in the material at the balls from their sample, and they found no significant impact. This characteristic analysis is critical to verify if UVC irradiation degrades the samples. Finally, Kim and Kang [72] found a synergistic effect from the combination of UVC treatment with mild temperature (60 °C).

### 2.7. Wavelengths and Fluence

After discussing the specificities for each scientific field, we move our attention to the tested wavelengths and the achieved intensity, described by the applied fluence (mJ/cm^2^), in each area. Given that UVC wavelengths fall within a range, future users must comprehend to what extent UVC LEDs have been applied in their specific area. Table 1 depicts the frequency that wavelengths were tested according to their scientific field.

The reader may identify that Table 1 summarizes more results than our SLR’s sample, encompassing 61 studies. Nevertheless, some researchers have evaluated more than one wavelength within the UVC range. 

There are some critical notes to remark. The first regards the most frequent combination between applied wavelengths and area, which lies in food studies using wavelengths from 270 nm to 279 nm. Even if this value would be inflated by the unclear text of Murashita et al. [59], who indicated the use of “[o]ne UVC-LED module (…) with wavelength of 270 to 280 nm” (p. 1199), it would still be the most frequent combination.

The second remark is quite surprising, given it indicates that researchers have been employing wavelengths that are neither the 260 nm (theoretical best), that reach the DNA/RNA absorbance spectrum [1,3,5,10,49,52] nor the 280 nm, capable of affecting proteins [3]. We did not find an explanation for this preference, but this might happen due to available LEDs for these researchers.

Thrid, two studies fail to state which wavelength(s) they used. Nevertheless, it was possible to determine that one had an effect below the disinfection level [63]. In contrast, the other had an unclear effect, as it did not evaluate the log-reduction of the biological indicators [40]. Finally, there is a clear opportunity for decontaminating materials and assessing this process’s effect on different materials. We propose this as a future research opportunity once the only study that followed this path was Trivellin et al. [70], who only assessed visual changes.

After analyzing the frequency results for wavelengths, we focus on fluence to explore the applied UVC intensity by researchers in different areas. We opted for fluence (mJ/cm^2^) instead of power density (mW/cm^2^) to mitigate the impact of missing data. Table 2 summarizes this information for each discussed area. Furthermore, we selected the maximum value for studies using multiple intensities, given that they always led to decontamination at some level.

From Table 2, we can regard a total (68) closer results to our SLR sample (61). This difference results from the employment of various fluencies for decontaminating biological indicators. The effect of missing values (NA) represents a significant concern for users willing to reproduce methods in the literature. This influence on research reproducibility can be mitigated if we consider that ten studies selected power density instead of fluence, reducing this number to 11 studies that fail to provide any intensity measure of their decontamination methods. Still, from Table 2, it is possible to verify the diversity of methods applied for water and food decontamination. Finally, future studies that assess material degradation have a broad set of possibilities if they wish to evaluate the impact of the UVC LEDs’ decontamination on the materials’ characteristics.

## 3. The Decontamination Efficacy of UVC LEDs on Different Biological Agents’ Species

This section presents the UVC LEDs’ decontamination capabilities in terms of different species of biological agents, which links to the second part of our objective.

### 3.1. Viruses

Viruses can have a single-stranded RNA (ssRNA), such as bacteriophage MS2 [5,7,19,47,48], bacteriophage Qβ [7,19], severe acute respiratory syndrome coronavirus 2 (SARS-CoV-2) [24,31,33,38,70], or other influenza viruses [3,32]. Alternatively, they have single-stranded DNA, like bacteriophage ΦX174 [7,19]. Viruses’ susceptibility to UVC depends on their genetic material; for instance, ΦX174 needs lesser irradiation (fluence) than MS2 or Qβ to reach virucidal effectiveness [7,19]. Another essential factor for the virucidal activity of UVC LEDs is the dosage received by the sample. The dose received varies according to the distance the samples are placed from the UVC sources [31] or irradiance received [47], among other variables presented in Table A1. 

Given ssRNA viruses are more difficult for UVC decontamination effectiveness, all studies in our sample use this group of viruses as biological indicators [3,5,7,19,31,32,33,38,47,48,70]. This genomic material comprises viral ribonucleoprotein complexes, which aggregate genome segments, RNA polymerases, and nucleoprotein [3,24]. UVC LEDs irradiation creates photochemical reactions, inhibiting the processes associated with ribonucleoprotein (transcription and translation) [3,5,24,48]. One must regard that UVC LED irradiation does not produce ROS that would lead to oxidation. Nor it directly affects the viral ribonucleoprotein [3,5], most likely for the lack of relative enzymes or cellular functionality [5,48]. 

The studies in our sample always showed some virucidal effectiveness. Only two studies could not disinfect their samples, finding results below two log-reduction [5,48]. In contrast, most studies [3,7,24,33,47,70] reached disinfection, and some studies reached disinfection or untraceable levels after some exposure periods [19,31,32,38]. Finally, some recent studies [33,47,70] applying SARS-CoV-2 or influenza virus as its surrogate frequently looked for untraceable results due to the pandemic’s context.

### 3.2. Bacteria

Bacteria represent a vast group of microorganisms with synergistic and harmful effects on humans. Bacteria can infect humans, causing different grievous diseases. This concern is ever more relevant since human actions and previous decontamination methods created drug-resistant mutations. Nevertheless, according to Dujowich et al. [18], UVC does not discriminate against drug-sensitive/-resistant microorganisms. Thus, it is an attractive option for users willing to reach decontamination.

Gram’s stain method divides bacteria into two groups: Gram-negative and Gram-positive. Our sample has representants of both types. Gram-negative bacteria were *Enterobacter* spp. [58], *Escherichia coli* [1,2,5,6,7,10,11,18,22,26,28,40,43,44,46,47,48,49,52,53,55,56,57,58,59,62,64,66,67,68,69,72], *Klebsiella pneumoniae* [23,34], *Legionella pneumophila* [32], *Pseudomonas aeruginosa* [21,22,28,29,30,34,36,50,57], *Pseudomonas alcaligenes* [69], *Pseudomonas brenneri* [54], *Raoultella ornithinolytica* [54], *Salmonella enteritidis* [18], *Salmonella senftenberg* [18], *Salmonella tennessee* [18], *Salmonella typhimurium* [1,6,7,34,43,45,51,56,59,61,62,63,67,68,69,72], *Serratia marcescens* [44,46,69], and *Vibrio cholerae* [57]. Whereas, Gram-positive bacteria were *Bacillus anthracis* [71], *Bacillus atrophaeus* [71], *Bacillus cereus* [56,60], *Bacillus pumilus* [60], *Bacillus subtilis* [10], *Bacillus spizizenii* [67], *Enterococcus faecalis* [22], *Enterococcus faecium* [23], *Listeria monocytogenes* [1,6,7,18,51,56,59,61,62,65,67,68,72], *Rothia mucilaginosa* [54], *Staphylococcus aureus* [7,18,22,23,28,34,36,37,40,54,56,65,66], *Staphylococcus epidermidis* [43,44,46,69], *Streptococcus mutans* [35,40], *Streptococcus sobrinus* [35]. 

A common approach is the comparison of the decontamination efficacy among representatives of both groups [1,6,7,10,18,22,23,28,34,36,40,43,44,46,51,54,56,59,61,62,66,67,68,69,72]. This comparison is important for final users, as they evaluate whether UVC displayed better decontamination rates at one group than at the other, which turned out to be true for Gram-negative for most studies [1,6,10,18,22,36,43,44,46,51,59,61,62,66,67,72]. Kim et al. [1] explain this result by the low appearance of UV products in Gram-positive bacteria. These results are justifiable by the thick peptidoglycan wall of Gram-positive bacteria [18,36,43,44,46] followed by a cytoplasmic lipid membrane [43]. Besides this protection, other aspects might increase this resistance, such as DNA repair ability or cell size [18,62]. 

Bacteria also have other repair mechanisms against decontamination methods, such as dark repair [2,26,52] and photoreactivation [2,52]. Both mechanisms allow bacteria to bypass the dimer changes created by UV irradiation. Dark (excision) repair consists of a light-independent replacement of damaged DNA [2,26,52]. Light repair (photoreactivation) uses light within 330–480 nm (near UVC) or visible light to activate the photolyase enzyme [2,52]. This enzyme binds to UV photoproducts to protect the DNA [2,52], this protection happens via the monomerization of “the cyclobutane ring of the pyr<>pyr” [2] p. 332. All the studies in our SLR found some decontamination levels emerging from UVC LEDs. Some reached disinfection levels [2,5,6,7,10,11,18,26,34,35,36,45,49,52,53,56,57,58,60,61,62,63,67,69,71], while a few achieved sterilization or untraceable levels [1,28,29,30,37,46,51,54,59,68]. 

### 3.3. Fungi

Several studies in the SLR analyzed the biocidal effect of UVC LEDs on fungi representatives, like *Alternaria japonica* [7], *Aspergillus flavus* [7], *Candida albicans* [34,40], *Pichia membranaefacien* [18], *Saccharomyces pastorianus* [18], and *Trichophyton rubrum* [39]. 

Some studies compare fungi decontamination using UVC LEDs with other biological agents. They suggest yeasts are far more resistant biological indicators than others [7,18,34]. According to Kim and Kang [7], fungi are more resistant because they represent eukaryotic cells. They are already more complex microorganisms than bacteria or viruses; thus, they hold different defense mechanisms. For instance, fungi use pigment production, nucleotide excision repair, and photoreactivation to defend themselves from UV irradiation [73]. According to Wong et al. [73], pigment production represents the first defense line for fungi, and the most common pigments are melanin, carotenoids, and mycosporines. Nucleotide excision repair, on the other hand, does incisions “on both sides of DNA” (p. 26) to remove photolesions [73], and this process applies almost 30 different proteins to be completed. 

This evidence indicates that for these biological indicators, UVC LEDs’ potential users must opt for higher dosages [7] or higher irradiation periods [34]. The results from our SLR indicate at least some decontamination levels [40]. Still, some found disinfection [7,18,34] or the ability to impede regrowth [39].

### 3.4. Other Biological Indicators

Only two studies discussed other biological indicators. One focused on using UVC LEDs to reduce mite infestation [41], discussed in Section 2.2. The other researchers analyzed the effect of UVC irradiation on plasmid vectors resistant to ampicillin and kanamycin [38]. Umar et al. [38] found a positive association between the size of the segment and the decontamination rates. Furthermore, these authors [38] were able to reach disinfection levels (≥4 log-reduction) using UVC at a fluence of 186 mJ/cm^2^.

### 3.5. Wavelengths and Fluence

After discussing the results for each biological indicator, we compare these results with the employed wavelengths and the irradiance intensity. Table 3 depicts the frequency that wavelengths had been tested according to their target biological indicator.

The analysis of the frequencies provided in Table 3 allows (prospective) users to determine whether the UVC LEDs are helpful for their disinfection systems. From this table, we also have some observations. The most common combination for researchers applying UVC LEDs tried to decontaminate bacteria using wavelengths within 270 to 279 nm. Additionally, the same remark goes to the possibility of this number being inflated, but even if the frequency is deflated, it remains the most recurrent. Although, in the previous subsections, it seemed that an extensive number of biological indicators had been assessed, there is a clear preference to evaluate bacteria over the other indicators. Only after COVID-19 pandemic viral indicators had gained importance in this analysis.

Shifting to the intensity of the decontamination method to eliminate biological indicators, we again select the method’s fluence over its power intensity to reduce missing values problems. Table 4 summarizes these results.

As in Table 2, there is a smaller value in the total (68), although it remains different from the SLR’s sample (61). Interestingly, the fluence selection for fungi does not agglomerate in the higher classes. We expected this preference, given they are more complex indicators owning multiple defenses that simple microorganisms did not. Nevertheless, this might happen because the decontamination of fungi is being assessed in vitro. When it happened in toenails (a much more complex surface due to its porosity), the selected higher fluence was 3200 mJ/cm^2^.

## 4. Comparison of UVC LEDs and Traditional Light Sources Characteristics

Since the Minamata Convention on Mercury, LEDs have gained particular attention in research and industrial applications [9]. This convention will produce a series of human and environmental safety benefits in the upcoming years, ranging from the phase-out of outdated processes to the end of primary mercury mining [9]. 

Researchers applying UVC in their experiments have been presenting various advantages LEDs have in comparison to mercury vapor light bulbs. First, LEDs do not have toxic substances, like mercury, representing an eco-friendlier option to traditional UVC emitting sources in terms of safety and waste management [1,2,5,6,7,10,19,22,26,28,32,47,48,49,53,54,55,62,63,68]. Second, in terms of operation, LEDs have longer service life [2,5,10,11,26,28,48,49,52,53,63,65,68] (approximately 100,000 h [11,52]) and they are easier controlled than conventional UVC sources [1,17,22,47,52,54,63,68], they do not need warm-up periods [1,2,6,7,28,47,49,53,56,65,68], nor they degrade over cycles [2,22,28,49,68]. Third, in terms of efficiency, LEDs emit low heat levels [1,6,7,19,28,62,63] and they have low energy consumption [2,5,11,17,22,28,48,55,56,63]. Fourth, from a physical perspective, LEDs are more robust [2,5,6,17,28,52,53,62] and compact [1,2,5,6,7,11,19,22,26,47,48,52,54,62,68], allowing their application into different systems designs. And finally, from an emission perspective, LEDs provide irradiance uniformity [11], multiple wavelengths [2,11,47,48,52,54,65], which can be combined to reach better decontamination effectiveness [5] or radiation power [22].

Nevertheless, traditional UVC sources, like mercury vapor lamps, also possess some advantages compared to LEDs. For instance, traditional UVC mercury vapor lamps have higher wall-plug efficiency (WPE, representing “the ratio of optical power output to electrical power input” p. 2 [47]) than LEDs [2,47,54], reaching almost 40% [54], while, currently, the diodes’ WPE only ranges from 1% to 3% [2]. Moreover, conventional UVC sources have higher irradiance efficiency than LEDs [19,47,52], which is crucial to reaching desired dosages [1]. In terms of cost, although some researchers advocate that LEDs are inexpensive [10], mercury vapor light bulbs are still cheaper than LEDs [52]. Table 5 compares the technical properties of UVC LEDs and mercury vapor lamps. 

From Table 5, the reader notes some of the technical properties that make LEDs so desirable compared to mercury vapor lamps. For example, LEDs’ service life in comparison to either low or medium-pressure vapor mercury lamps. Furthermore, while mercury vapor lamps need a warm-up period to reach their peak irradiance, LEDs do not. The only point that could create doubts in users about LEDs is their WPE. Nevertheless, with technological advances, it is expected that LEDs’ WPE will reach the 10% threshold by 2022 [13].

Regarding the decontamination effectiveness, LEDs can emit wavelengths of 260 nm, reaching the DNA/RNA absorbance spectrum [1,3,5,10,49,52], or 280 nm, capable of affecting proteins [3], while conventional UVC sources peak at 254 nm [5,49]. In our SLR, several researchers [1,19,23,44,46,59,67,71] compared the decontamination capabilities of mercury vapor lamps and LEDs, finding that LEDs had at least as good results as traditional sources in practically every scenario (Table 2). There are some exemptions, though, for instance: (1) Nunayon et al. [46] found that LEDs only reduced 1-log of *S. epidermis* concentration, while mercury vapor lamps achieved a 4.2-log reduction; (2) Wood et al. [71] report better results for traditional UVC sources in high relative humidity scenarios. Table 2 compares the decontamination efficacy between UVC LEDs and mercury vapor lamps. Given Table 6 is a subset of Table A1, we only provide the best results in each case.

Some authors discuss the possibility of LEDs having better results than mercury lamps. These better results are likely to come from LEDs’ convergent irradiation, while mercury vapor lamps have more dispersed irradiation. Thus, despite mercury vapor lamps having more irradiance intensity, they tend to have not as good results in terms of decontamination efficacy compared to LEDs [1]. 

Finally, one must regard that UVC decontamination may face potential problems despite its source. Porous materials might impact the decontamination efficacy of UVC irradiation, no matter the source [72]. For UVC decontamination to happen, its surfaces and materials must be irradiated. However, UVC irradiation might be hindered when dealing with complex materials, given that they might be porous or have demanding topologies. Porous materials may pose a penetration problem to UVC decontamination.

In contrast, topologically complex materials may create a shadowing effect, impeding the complete irradiation, creating areas where microorganisms can escape from UVC irradiation [16]. Once UVC decontamination might face problems, some researchers tried to couple this decontamination method with other possibilities. For readers interested in these combinations, see Appendix B. 

## 5. Conclusions

Academia and Industry have been applying LEDs to decontamination systems. These light sources represent a practical alternative to traditional light sources since they are more efficient, robust, compact, flexible, eco-friendlier, and have an extended lifecycle. Besides these advantages, users have been looking for alternatives since most countries became parties in the Minamata Convention on Mercury. Furthermore, this agreement might reduce the market for products containing mercury worldwide. 

Among the different wavelengths LEDs can provide, UVC has the potential to become the prime choice among users, as it has decontamination capabilities. Such feature is so well-established that the FDA (the US Food and Drug Administration) has allowed its application in food, water, and beverages treatments since 2000 [6,19].

This research used an SLR for three objectives. First, we divided the different studies according to their areas, presenting the current state of the art in each one and dividing the results in terms of decontamination effectiveness. At this point, we classified the findings of each research in terms of disinfection and sterilization levels. The UVC LED’s decontamination (germicidal) effectiveness is a fact. However, we noticed that the more complex the sample became, the lower the researchers’ results. Most likely, this is due to attenuation. This problem is discussed in Optics by the empirical relation stated in the Bouguer-Beer-Lambert law, which relates light attenuation with the irradiated material properties. To overcome this problem, users may apply higher dosages, changes in the system geometry, or combine the UVC method with other decontamination methods.

Second, we divided the results in terms of biological agents. From the SLR results, all the studies indicate some decontamination levels. However, UVC LEDs are more effective for some biological representatives than others, even within the same species. From a broader perspective, the more complex the biological indicator is, the more challenging it is to decontaminate it from a specific medium. This consideration makes sense as more complex targets have more defense mechanisms against UVC products. Furthermore, they do not depend on just one cell holding all its genetic code.

This research’s third objective was to enlighten whether UVC LEDs were suitable alternatives for conventional UVC light sources, like mercury vapor lamps, in terms of decontamination levels. This analysis could summarize a wide range of advantages LEDs compared to traditional light sources. Still, they remain an expensive choice and have low WPE. Furthermore, several studies compare these light sources’ decontamination capabilities, finding that UVC LEDs are as good as mercury vapor lamps in this criterion.

Our results are crucial for future researchers for multiple reasons. First, we notice that UVC LEDs can still be applied to decontaminate numerous targets that were not evaluated. This observation is cardinal for industrial applications given the pandemics we are still facing, providing avenues for future research. Second, users must also account for possible material changes after irradiation. As previously discussed, complex targets might need higher dosages, which can degrade surfaces. Third, these results might incentivize LED producers to improve their products and become even more cost-effective than traditional light sources. Fourth, our results provide potential users some assurance for those willing to do a smooth transition from their decontamination systems to alternative light sources, especially in the upcoming years with the obligations held by their countries in the Minamata Convention.

This research has one major limitation: our conclusions are built upon other researchers’ findings. Ergo, our findings are directly connected to the authors’ conclusions presented in the SLR provided, which could have been misinformed or could not have been clear.

## Figures and Tables

**Figure 1 materials-15-02854-f001:**
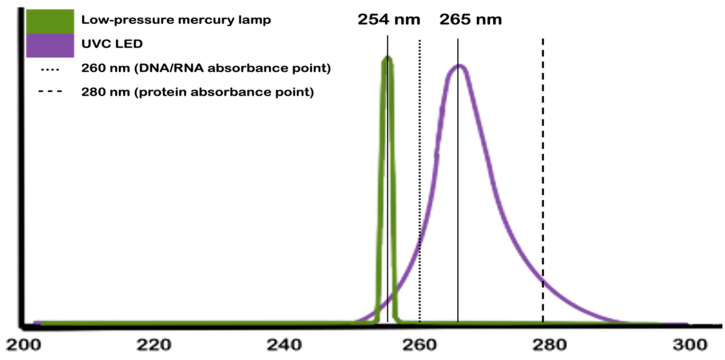
Wavelength peaks according to the UVC sources. Note: We abstained from providing the curve for medium-pressure lamps because they overlap with one of the UVC LEDs. For technical parameters about UVC traditional sources and LEDs, we suggest moving to Section 4.

**Figure 2 materials-15-02854-f002:**
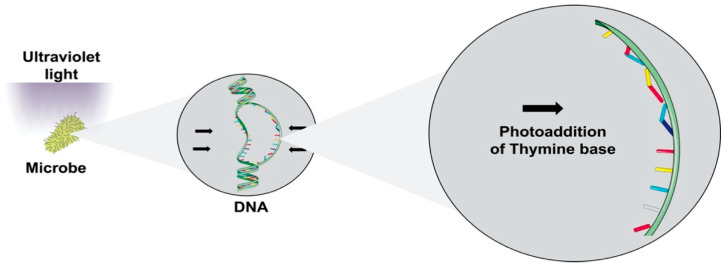
Photodimerization process. Source: Smart Service Medical Art.

**Table 1 materials-15-02854-t001:** Used wavelengths according to the scientific field.

Wavelength (nm)	Area						
	Air	Food	Health	Materials	Veterinary	Water	Total
Below 260	0	0	2	0	0	1	3
From 260 to 269	1	3	11	2	0	12	29
From 270 to 279	3	25 ^1^ (17)	7	1	2	10	48 (40)
280	3	4	4	0	1	4	16
NA	0	2	1	0	0	0	3
Total	7	33 (25)	26	3	3	27	99 (91)

Notes: “NA” indicates that the research was unclear on the employed wavelength(s). ^1^ This value may be inflated by the ambiguous text from Murashita et al. [59], as the authors only indicate they used LEDs that could range from 270 nm to 280 nm but did not clearly state which wavelength they employed. For comparison purposes, we deflated all the numbers left in parentheses. We considered only one wavelength within the 270 to 279 nm range to deflate it.

**Table 2 materials-15-02854-t002:** Fluencies according to the scientific field.

Fluence (mJ/cm^2^)	Area						
	Air	Food	Health	Materials	Veterinary	Water	Total
0 to less than 10	1	5	0	2	0	7	15
10 to less than 100	2	2	7	0	0	6	17
100 to less than 1000	0	2	2	0	0	3	7
at least 1000	0	2	4	1	0	1	8
NA	4	3	8	0	1	5	21
Total	7	14	21	3	1	22	68

Notes: “NA” indicates that the research was unclear on the employed fluence during their decontamination process.

**Table 3 materials-15-02854-t003:** Wavelengths according to the decontaminated biological indicators.

Wavelengths (nm)	Biological Indicators				
	Bacteria	Fungi	Others	Viruses	Total
Below 260	2	0	0	1	3
From 260 to 269	20	2	1	6	29
From 270 to 279	39 ^1^ (31)	3	2	4	48 (40)
280	9	2	1	4	16
NA	2	1	0	0	3
Total	72 (64)	8	4	15	99 (91)

Notes: “NA” indicates that the research was unclear on the employed wavelength(s). ^1^ This value may be inflated by the ambiguous text from Murashita et al. [59], as the authors only indicate they used LEDs that could range from 270 nm to 280 nm but did not clearly state which wavelength they employed. For comparison purposes, we deflated all the numbers left in parentheses. We considered only one wavelength within the 270 to 279 nm range to deflate it.

**Table 4 materials-15-02854-t004:** Fluencies according to the decontaminated biological indicators.

Fluence (mJ/cm^2^)	Biological Indicators				
	Bacteria	Fungi	Others	Viruses	Total
0 to less than 10	12	1	0	2	15
10 to less than 100	11	1	0	5	17
100 to less than 1000	4	0	1	2	7
at least 1000	6	1	0	1	8
NA	17	2	1	1	21
Total	50	5	2	11	68

Notes: “NA” indicates that the research was unclear on the employed wavelength(s).

**Table 5 materials-15-02854-t005:** A comparison of the technical properties of UVC LEDs and mercury lamps.

Variable (Unit)	UVC Source		
	LEDs	Low-Pressure Mercury Vapor Lamps	Medium-Pressure Mercury Vapor Lamps
UVC spectral width (nm)	200–280 [52]	254 [5]	200–280 [14]
Service life (h)	Up to 100,000 [52]	8000–12,000 [14]	4000–8000 [14]
Wall-plug efficiency (%)	1–3 [2]	Up to 40 [54]	10–20 [14]
Cold start time (min)	negligible [2]	4–7 [14]	1–5 [14]
Warm start time (min)	negligible [2]	2–7 [14]	4–10 [14]

**Table 6 materials-15-02854-t006:** A comparison of the decontamination efficacy between UVC LEDs and mercury vapor lamps.

Area	Biological Indicator	LEDs	Mercury Vapor Lamps	Source
Food	Bacteria	Sterilization (~6, *S. typhimurium)*	Disinfection (~3, *S. typhimurium*)	[1]
Food	Bacteria	Sterilization (>6, *E. Coli)*	Sterilization (>6, *E. Coli)*	[59] ^1^
Food	Bacteria	Disinfection (>4, *S. typhimurium*)	Disinfection (>4, *S. typhimurium*)	[67] ^2^
Health	Bacteria	Disinfection, “There were no significant changes (…) between samples (…) treated with different light sources” (p. 1)		[23]
Air	Bacteria	1.148 m^2^/J (*S. marcens*)	0.042 m^2^/J (*S. marcens*)	[44] ^3^
Air	Bacteria	Sterilization (>7.4, *E. Coli)*	Sterilization (>7.1, *E. Coli)*	[46]
Materials	Bacteria	Disinfection (5.06, *B. atrophaeus*)	Disinfection (4.73, *B. anthracis*)	[72]
Water	Viruses	Sterilization (>6, ΦX 174)	Disinfection (~4, ΦX 174)	[19]

Notes: ~ indicates approximately. ^1^ Although both cases reached sterilization, mercury vapor lamps required a more extended period. ^2^ The authors only compared the sources in the disinfection of a plastic surface. ^3^ The authors only evaluate the biological indicators susceptibility to UVC after each source exposure.

## Data Availability

The SLR protocol used to reproduce our results is presented in Appendix A. The final database we derived from the SLR, which we used to describe and discuss our results.

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
