# Peer review of "A Comprehensive Analysis of the UVC LEDs’ Applications and Decontamination Capability"

_materials, 2022, doi:10.3390/ma15082854_

Round 1

Reviewer 1 Report

The article concerns the effectiveness of decontamination processes carried out with light-emitting diodes operating in the UVC spectral range. It is of a review character, not a presentation of original research, so it is not understandable to use the terms "observed observed". It refers to known photonic components not materials so it does not fit the scope of the journal which is as follows:

  • To publish research related to all classes of materials including ceramics, glasses, polymers (plastics), composites, semiconductors, magnetic materials, biological and biomimetics materials, silica, dots, and carbon materials, metals, and alloys from nanoscale to bulk. All kinds of functional materials used for the development of medical implants in medicine and in dentistry, coatings and films, pigments, ionic crystals, covalent crystals, metals, and intermetallics are also considered.
  • To cover all aspects of materials science or materials engineering, including nanoscience and nanotechnology.
  • To contribute to the advancement of material characterization techniques such as electron microscopy, X-ray diffraction, calorimetry, nuclear microscopy and spectroscopy, laser technology, optical fibers, Rutherford backscattering, and neutron diffraction, among others.
  • To incite fundamental research in condensed matter physics and materials physics, continuum mechanics and statistics, mechanics of materials, tribology (friction, lubrication and wear), solid-state physics, etc.

Subject of the paper is not in agreement with keywords of the Special Issue which  is “Design and fabrication of:

- Materials for active optical elements

- Materials for passive optical elements

- Bioinspired materials

- Biomimicking materials

- Nanoparticles for imaging and sensing”

The authors made an extensive review of the experiences of other teams with decontamination with UVC diodes. It should be noted, however, that they limited themselves to listing the works and assigning them to the area of application, without, however, making an in-depth comparative analysis that would provide useful information for the user. There is no quantitative comparison with discharge lamps (illumination, spectral width, exposure time), standards and commercial UVC products used crosswise for the same purposes. Also, individual diode solutions have been tabulated, but it is difficult to read on the basis of the table which experiments have brought satisfactory decontamination according to the applicable standards and which have a real benefit in relation to the existing solutions.

Author Response

Reviewer 1

"The article concerns the effectiveness of decontamination processes carried out with light-emitting diodes operating in the UVC spectral range. It is of a review character, not a presentation of original research, so it is not understandable to use the terms "observed observed". It refers to known photonic components not materials so it does not fit the scope of the journal which is as follows:

  • To publish research related to all classes of materials including ceramics, glasses, polymers (plastics), composites, semiconductors, magnetic materials, biological and biomimetics materials, silica, dots, and carbon materials, metals, and alloys from nanoscale to bulk. All kinds of functional materials used for the development of medical implants in medicine and in dentistry, coatings and films, pigments, ionic crystals, covalent crystals, metals, and intermetallics are also considered.
  • To cover all aspects of materials science or materials engineering, including nanoscience and nanotechnology.
  • To contribute to the advancement of material characterization techniques such as electron microscopy, X-ray diffraction, calorimetry, nuclear microscopy and spectroscopy, laser technology, optical fibers, Rutherford backscattering, and neutron diffraction, among others.
  • To incite fundamental research in condensed matter physics and materials physics, continuum mechanics and statistics, mechanics of materials, tribology (friction, lubrication and wear), solid-state physics, etc.

Subject of the paper is not in agreement with keywords of the Special Issue which  is "Design and fabrication of:

- Materials for active optical elements

- Materials for passive optical elements

- Bioinspired materials

- Biomimicking materials

- Nanoparticles for imaging and sensing"

The authors made an extensive review of the experiences of other teams with decontamination with UVC diodes. It should be noted, however, that they limited themselves to listing the works and assigning them to the area of application, without, however, making an in-depth comparative analysis that would provide useful information for the user. There is no quantitative comparison with discharge lamps (illumination, spectral width, exposure time), standards and commercial UVC products used crosswise for the same purposes. Also, individual diode solutions have been tabulated, but it is difficult to read on the basis of the table which experiments have brought satisfactory decontamination according to the applicable standards and which have a real benefit in relation to the existing solutions."

R: We appreciate the time for reviewing our research. Please, read below our answers and suggestions.

"It refers to known photonic components not materials, so it does not fit the scope of the journal"

R: We respectfully disagree with this statement. As indicated by Reviewer 1, part of the Journals' scope lies in: "To cover all aspects of materials science…", and the decontamination process is mandatory in a broad scope of material applications and may significantly affect material's performance. Therefore, in our opinion, it falls within the scope of the Materials journal. Unfortunately, only one study (Trivellin et al. [72]) evaluated visual changes in their decontaminated samples within our extensive literature review. We speculate that they would have done further analyses regarding material characterization if they had found significant differences.

Our research highlights the importance "(…) the increase in the targets' complexity (e.g., multilayers or thicker individual layer) may reduce the UVC decontamination efficacy. Therefore, UVC LEDs still require considerable optimization. These findings are stimulating for developing industrial or final users' applications." In our opinion, these are crucial features that Material science researchers must not disregard.

"It is of a review character, not a presentation of original research, so it is not understandable to use the terms "observed observed"."

R: We have used this verb in its intransitive form, meaning "remark, comment." Nevertheless, we understand the Reviewers' point of view. Thus, we exchanged this term with other verbs to mitigate possible misunderstandings.

"The authors made an extensive review of the experiences of other teams with decontamination with UVC diodes. It should be noted, however, that they limited themselves to listing the works and assigning them to the area of application, without, however, making an in-depth comparative analysis that would provide useful information for the user. There is no quantitative comparison with discharge lamps (illumination, spectral width, exposure time), standards and commercial UVC products used crosswise for the same purposes."

R: As requested, an in-depth comparative analysis of the technical properties and the decontamination efficacy between UVC LEDs and mercury lamps were added in Section 4 and added two tables (tables 5 and 6)

"Also, individual diode solutions have been tabulated, but it is difficult to read on the basis of the table which experiments have brought satisfactory decontamination according to the applicable standards and which have a real benefit in relation to the existing solutions."

R: Table A1 was reorganized according to the best results described. In addition, it shows in dark grey the sterilization or untraceable levels. Then it depicts results in light grey representing the disinfection results. Finally, the uncolored cases represent results beneath the decontamination threshold.

Reviewer 2 Report

The manuscript written by Talita Nicolau comprehensively analyzed the application and decontamination ability of UVC LED and compared it with mercury lamp. This study observed the application of UVC LED in different areas, and roughly compared the decontamination effect of UVC LED and conventional lights. However, there is not enough comparative data to support this view. The work is not attractive enough to the reader. I would suggest a major revision before publication consideration.

  1. In order to get the point across, more information on the comparison between mercury lamp and UVC LED should be added in the Introduction part.
  2. In the fourth paragraph of the Introduction part, the author narrows the decontamination concept down into two additional concepts (disinfection and sterilization), but does not discuss the relationship between the additional concepts and UVC LED.
  3. In the Section 2 and Section 3, the author presents the application of UVC LEDs in distinct areas. But, there are no discussion of the decontamination effectiveness compared with the traditional mercury lamp.

Author Response

Reviewer 2

"The manuscript written by Talita Nicolau comprehensively analyzed the application and decontamination ability of UVC LED and compared it with mercury lamp. This study observed the application of UVC LED in different areas, and roughly compared the decontamination effect of UVC LED and conventional lights. However, there is not enough comparative data to support this view."

R: We want to thank you, the reviewer, for reviewing our research. We added Tables 5 and 6 to summarize the technical properties and decontamination efficacy between UVC LEDs and mercury lamps. We remain within our SLR's sample to write each section, which might have been the reason for the lack of comparative data. Only eight studies out of the 61 compare UVC LEDs and traditional sources. All studies focused on mercury vapor lamps limitations, but usually in a brief qualitative approach. The referred studies primarily focus on reducing biological indicators (CFU/ml or PFU/ml).

"1. In order to get the point across, more information on the comparison between mercury lamp and UVC LED should be added in the Introduction part."

R: We agree. The requested comparison was added in the paragraph starting at line 38.

"2. In the fourth paragraph of the Introduction part, the author narrows the decontamination concept down into two additional concepts (disinfection and sterilization) but does not discuss the relationship between the additional concepts and UVC LED."

R: To meet this request, we added this relationship in the paragraph starting at line 74.

"3. In the Section 2 and Section 3, the author presents the application of UVC LEDs in distinct areas. But there are no discussion of the decontamination effectiveness compared with the traditional mercury lamp."

R: The authors added information comprising mercury vapor lamp performance. This information is depicted in a new table (Table 6) and critically discussed in Section 4.

Reviewer 3 Report

The manuscript entitled “A comprehensive analysis of the UVC LEDs’ applications and decontamination capability” by Talita et al. has reviewed the recent process of UVC LED in the aspect of decontamination. While the proposed review is meaningful in this field, the current manuscript can be improved after the following revisions.

  1. The major concern from the reviewer is that the decontamination capability of UVC LED could be discussed more in the text in the comparison with other methods, which can be also presented as a table form.

  1. In section 2.2. Animal health, the authors stated that “Only one study concentrates on Veterinary issues”, which may be not correct. The authors may check whether there are more studies on Veterinary issues such as aquaculture.

  1. The limitations/disadvantages of UVC LEDs decontamination should be at least briefly introduced in the text.

  1. Some of the expressions are not very clear, such as “Considering decontamination effectiveness, two studies partially disinfected biological indicators, and one study partially sterilized the analyzed indicators” in section 2.3. Air treatment and “Although few studies [5,18,43,44,46,52] were not able to reach disinfection levels, the majority [2,22,43,45,48– 186, 50,53–55,57] reached them” in 2.4. Water treatment.

Minor

Some formatting errors were found such as “Chemical notations: O3, ozone. H2O2, hydrogen peroxide” in Notes at the end of Appendix A. The authors should check the overall text for accuracy.

Author Response

Reviewer 3

"The manuscript entitled "A comprehensive analysis of the UVC LEDs' applications and decontamination capability" by Talita et al. has reviewed the recent process of UVC LED in the aspect of decontamination. While the proposed review is meaningful in this field, the current manuscript can be improved after the following revisions."

R: We appreciate the time for reviewing our research. Please, read below our answers and suggestions.

"1. The major concern from the reviewer is that the decontamination capability of UVC LED could be discussed more in the text in the comparison with other methods, which can be also presented as a table form."

R: We agree. To meet this requirement, we added Appendix B.

"2. In section 2.2. Animal health, the authors stated that "Only one study concentrates on Veterinary issues", which may be not correct. The authors may check whether there are more studies on Veterinary issues such as aquaculture."

R: We clarified in section 2.2 that this limitation relates to our SLR's database and not the area itself. In addition, we also provided a reference for readers interested in this scientific field.

"3. The limitations/disadvantages of UVC LEDs decontamination should be at least briefly introduced in the text."

R: We added this discussion in the last paragraph of section 4.

"4. Some of the expressions are not very clear, such as "Considering decontamination effectiveness, two studies partially disinfected biological indicators, and one study partially sterilized the analyzed indicators" in section 2.3. Air treatment and “Although few studies [5,18,43,44,46,52] were not able to reach disinfection levels, the majority [2,22,43,45,48– 186, 50,53–55,57] reached them” in 2.4. Water treatment."

R: We addressed these vague statements. Please, see the yellow marks in lines 180 and 225, respectively.

"Some formatting errors were found such as "Chemical notations: O3, ozone. H2O2, hydrogen peroxide" in Notes at the end of Appendix A. The authors should check the overall text for accuracy."

R: Thank you for such a thorough review. We addressed this problem in line 664.

Reviewer 4 Report

The authors did a comprehensive review on the application of UVC LED for decontamination in human health, air treatment, water treatment and food treatment etc. The application field and the comparison to conventional mercury lamps were discussed. This manuscript is well written and it is interesting to the readers. It can be published in Materials after addressing the following questions:

1. The specific wavelength and intensity has to be summarized or discussed for treatment of different objects and application fields since UVC is a range and the wavelength and intensity will affect the efficiency.

2. The recently developed treatment of COVID-19 in aerosol state is suggested to be included in 2.3.

3. There is only one figure in this manuscript. More figures of the treatment strategy are suggested to be included for better understanding, either from the literatures or draw by the authors.

Author Response

Reviewer 4

"The authors did a comprehensive review on the application of UVC LED for decontamination in human health, air treatment, water treatment and food treatment etc. The application field and the comparison to conventional mercury lamps were discussed. This manuscript is well written and it is interesting to the readers. It can be published in Materials after addressing the following questions:

"1. The specific wavelength and intensity has to be summarized or discussed for treatment of different objects and application fields since UVC is a range and the wavelength and intensity will affect the efficiency."

R: We would like to acknowledge the reviewer for carefully revising our research. Our analysis depicted two standard intensity variables (fluence, mJ/cm², and power density, mW/cm²). Thus, we opted for addressing intensity using fluence, primarily to mitigate the effect of missing data. We have added two new sections to meet the reviewer's request (Sections 2.7 and 3.5).

"2. The recently developed treatment of COVID-19 in aerosol state is suggested to be included in 2.3."

R: As suggested, information concerning this treatment has been added in Section 2.3, line 187.

"3. There is only one figure in this manuscript. More figures of the treatment strategy are suggested to be included for better understanding, either from the literatures or draw by the authors."

R: An additional Figure (Figure 1) and a Graphical Abstract that illustrates various aspects of our paper were added to the manuscript.

Round 2

Reviewer 1 Report

The article concerns the effectiveness of decontamination processes carried out with light-emitting diodes operating in the UVC spectral range. It is of a review character, not a presentation of original research. It refers more to photonic components than to materials, however  after explanation of the Authors and improvement of the manuscript which pointed interaction of light and different illuminated materials, I think that paper sufficiently fits the scope of the journal and special issue. Improvement made by Authors addressed my other comments. Content of the paper is sufficient to its form of reviewing the state of knowledge and technology. Summarising, in present form the paper can be recommended for publication.

Author Response

We appreciate the time for reviewing our research.

Reviewer 2 Report

The manuscript written by Talita Nicolau comprehensively analyzed the application and decontamination ability of UVC LED and compared it with mercury lamp. This study observed the application of UVC LED in different areas, and roughly compared the decontamination effect of UVC LED and conventional lights. However, there is not enough comparative data to support this view. The work is not attractive enough to the reader. I would suggest a major revision before publication consideration.

  1. In order to get the point across, more information on the comparison between mercury lamp and UVC LED should be added in the Introduction part.
  2. In the fourth paragraph of the Introduction part, the author narrows the decontamination concept down into two additional concepts (disinfection and sterilization), but does not discuss the relationship between the additional concepts and UVC LED.
  3. In the Section 2 and Section 3, the author presents the application of UVC LEDs in distinct areas. But, there are no discussion of the decontamination effectiveness compared with the traditional mercury lamp.

Author Response

We believe that there might have been some mistake in the report of Reviewer 2. The review report is an exact copy of the first round of reviews. We would like to state that all these comments were addressed in the first round of revisions.